# Sterilizing Ready-to-Eat Poached Spicy Pork Slices Using a New Device: Combined Radio Frequency Energy and Superheated Water

**DOI:** 10.3390/foods11182841

**Published:** 2022-09-14

**Authors:** Ke Wang, Chuanyang Ran, Baozhong Cui, Yanan Sun, Hongfei Fu, Xiangwei Chen, Yequn Wang, Yunyang Wang

**Affiliations:** College of Food Science and Engineering, Northwest A&F University, Yangling 712100, China

**Keywords:** RF heating, superheated water, poached spicy pork slices, sterilization, quality, RTE food, *G. stearothermophilus* spores

## Abstract

In this study, a new device was used to inactivate *G. stearothermophilus* spores in ready-to-eat (RTE) poached spicy pork slices (PSPS) applying radio frequency (RF) energy (27.12 MHz, 6 kW) and superheated water (SW) simultaneously. The cold spot in the PSPS sample was determined. The effects of electrode gap and SW temperature on heating rate, spore inactivation, physiochemical properties (water loss, texture, and oxidation), sensory properties, and SEM of samples were investigated. The cold spot lies in the geometric center of the soup. The heating rate increased with increasing electrode gap and hit a peak under 190 mm. Radio frequency combined superheated water (RFSW) sterilization greatly decreased the come-up time (CUT) compared with SW sterilization, and a 5 log reduction in *G. stearothermophilus* spores was achieved. RFSW sterilization under 170 mm electrode gap reduced the water loss, thermal damage of texture, oxidation, and tissues and cells of the sample, and kept a better sensory evaluation. RFSW sterilization has great potential in solid or semisolid food processing engineering.

## 1. Introduction

Ready-to-eat (RTE) food, also known as convenient and precooked food, is earning the attention of people now with increasing income, for it is time-saving, hassle-free, and convenient [1]. Numerous research works have focused on safety control and new processing technologies of RTE food, but the products purchased by consumers are still at risk of contamination by microbes, such as *Listeria*, *Escherichia coli*, *Enterobacter sakazakii*, etc. [2,3,4].

Poached spicy pork slices (PSPS), a traditional Chinese cuisine available in any Sichuan restaurant worldwide, might have a considerable market if it was developed into an RTE food. Like other RTE meat products, RTE PSPS need to be sterilized as fore-cooking contamination may occur when PSPS are exposed to air, equipment, or food handlers [5]. Theoretically, heating of meat or meat product would induce protein degradation, myofibrillar shrinkage, and consequently water loss, causing tremendous changes in texture, color, sensory properties, and so on [6,7]. To ensure that the internal temperature of the product reaches the required temperature, high-pressure steam/water sterilization at 121 °C for 30 min is commonly used in industry, which will also cause the above-mentioned changes. The intrinsic quality of a meat product is an essential factor affecting consumer preference and the degree of acceptability for consumers [8]. Therefore, it is necessary to develop an innovative sterilization method to decrease the sterilization time while maximizing the preservation of its texture, color, and nutrition.

Radio frequency (RF) is an electromagnetic wave with a frequency ranging from 3 kHz to 300 MHz. RF heats food material by driving the polar molecules’ rotation and movement of charged ions. RF heating has been studied in the drying, blanching, and sterilization of food materials because of its advantages of large penetration depth and high heating efficiency [9,10,11,12]. Nonuniform heating is the main obstacle to the industrialization of RF. Various kinds of food materials have been studied, such as meat products [13], grain products [14,15], and vegetable products [16], but reports about RTE food are rare.

Radio frequency combined hot water (RFHW) heating is not a new idea. Some previous studies carried out RFHW with two steps of RF preheating and a hot water bath, discovering it can greatly shorten the heating time and improve the quality of the product. Yang and Geveke’s [17] study showed that RFHW heating can shorten the sterilization time by 78% while reducing the *salmonella* in eggs by at least 5 log and causing unobservable quality change. Xu et al. [14] successively applied a hot water bath and RF to *Nostoc sphaeroides*, observing that its temperature uniformity, nutrient content, and phycocyanin stability were different compared with samples subjected to high-pressure steam sterilization.

Although there have been many studies on RFHW heating, the literature on how the superposition of the two heating methods simultaneously affect the sterilization results and quality is scarce. Our laboratory designed a device that can carry out radio frequency combined superheated water (RFSW) sterilization, which overcomes this obstacle in terms of equipment. This study aimed to do the following: (1) determine the cold spot in PSPS during RFSW heating; (2) investigate the effect of superheated water (SW) temperature and electrode gap on heating rate and inactivation of spores of *G. stearothermophilus*; and (3) compare the weight loss, texture, and sensory and morphological structure differences between sterilization methods of RFSW and SW alone.

## 2. Materials and Methods

### 2.1. PSPS Preparation

The cooking method of PSPS in this article referred to the traditional Chinese method with moderate modification. Cooking-related material such as seasonings and pork tenderloin were purchased from the Hao&Duo supermarket in Yangling, China. Spring onion, ginger, and garlic were cut into pieces. In a wok, edible oil was heated to 150 °C.

Then, spring onion, ginger, garlic, and bean paste were added and stirred constantly for 40 s. Water was poured in, and the prepared seasonings were added. After boiling, the samples were removed, cooled, and stored at 4 °C for later use within 2 days. Pork tenderloin was cut into 1.0 cm × 3.0 cm × 1.0 cm slices (length × width × thickness) and placed into a vessel. Salt, vinegar, egg white, and potato starch were added successively and mixed adequately, and then the meat was left to stand for 30 min. The pork slices were uncrumpled, and then boiled for 4 min. They were removed with a colander, placed into a sterile vessel, and stored at 4 °C for later use within 2 days. The amount of all materials is shown in Table 1.

The sample (51.00 ± 0.50 g; soup 38.50 ± 0.04 g and pork slices 12.80 ± 0.03 g) was weighed and placed in a Teflon container (Figure 1). The Teflon container was sterilized by ultraviolet radiation prior to use. The ratio of height to diameter of the material was kept at 2.2. To keep the same initial temperature, the sample was held at 25 °C for 2–4 h before heating.

A waterproof and high-temperature-resistant adhesive was used to stick a plastic nut to the selected position outside the Teflon container. The plastic nut provided a cone thread, so that the fiber optic could be squeezed to isolate water when the plastic screw was tightened inward. The PSPS was inevitably stratified due to gravity. The pork slices gathered in the bottom of the container, and edible oil floated on the soup (Figure 1).

### 2.2. Bacterial Culture and Spore Crop Preparation

The bacterial culture and spore crop preparation of this study was based on the method of Ahn et al. [18] with moderate modification. A strain of *G. stearothermophilus* (ATCC 7953, Shanghai Fuxiang Biotechnology Co., Ltd., Shanghai, China) was cultivated aerobically in a nutrient broth (NB, AOBOX biotechnology, Beijing, China) at 55 °C for 48 h. NB (200 μL) was evenly plated on the surface of NA (Land bridge, Beijing, China) containing 50 ppm MnSO_4_ (Ghtech, Guangdong, China) and cultured at 55 °C for 5-10 days.

For bacteria suspension collection, 10 mL distilled water was added to each culture medium prior to scraping and pouring into a centrifuge tube. The bacteria suspension was heated at 90 °C for 30 min to inactivate the vegetative cells. Then, they were washed thrice by 6000× *g* centrifugation for 5 min. The collected spore suspension was resuspended in deionized water to 10^7^–10^8^ (CFU mL^−1^) and stored at 4 °C for later use within 2 months.

### 2.3. RFSW Heating System

The RFSW heating system consisted of four parts of the SW generator (Figure 2a), an RF system (Figure 2b), sterilization kettle, and fiber optic temperature measurement system (Figure 2c). The SW generator with a power of 10 KW (TW-SP-25; Shanghai Triowin Automation Machinery Co., Ltd., Shanghai, China) could produce SW up to 130 °C and could automatically control the pressure in the sterilization kettle. The SW generator provided on-line cooling and it stopped when the internal temperature of the samples in the kettle reached the preset. A 6 KW, 27.12 MHz free running RF system (GJG-2.1-10AJY; Hebei Huashijiyuan High Frequency Equipment Co., Ltd., Hebei, China) with two parallel electrodes was used in this study. The electrode gap of the RF system ranged from 100 mm to 300 mm. The schematic of the sterilization kettle designed by our lab is shown in Figure 3. The sterilization kettle was composed of a hollow cylindrical cavity, two water pipes, and one pipe used for the fiber optic senor. For sealing, the top and bottom lids were fixed with the cavity by screws. The fiber optic temperature measurement system (HQ-FTS-D1F00, Herch Opto Electronic Technology Co., Ltd., Xi’an, China) consisted of fiber optic sensors, signal converter, and computer software. Detailed information is described in the studies by Wang et al. [19], Cui et al. [20], and Sun et al. [21].

In preparation of the experiment, the SW generator was connected to the sterilization kettle with a thermal insulation water pipe. The Teflon container containing samples was positioned at the center of the bottom lid of the sterilization kettle. The fiber optic sensors were inserted into the sample at designated points (points A to H in Figure 1). Then, the top lid was tightened with bolts. The sterilization kettle was placed on the bottom electrode in the RF cavity (Figure 2b). The electrode gap was adjusted according to the test arrangement. Then, SW was pumped into the kettle, and the RF heating system was started. For safety reasons, the pressure in the kettle was maintained between 0.2 and 0.3 bar automatically. In particular, SW flowed in from the bottom of the kettle and flowed out from the upper part to fill the kettle with SW. The Teflon container containing samples was immersed in SW. The on-line cooling procedure was started immediately when the designated sterilization time was reached. Then, the SW was closed when the temperature in the kettle went down to 40 °C. This is because meat products should be cooled rapidly after heating treatment, since overheating destroys the quality of meat. By this structure, the sample can be heated by SW and RF simultaneously.

### 2.4. The Cold Spot and Heating Rate

To explore the cold spot of the sample in the Teflon container under RFHW heating, the fiber optic sensor was inserted at eight points distributed in three layers of PSPS. Two points (E and H) were inside the pork slices (Figure 1). The device was assembled as described in Section 2.3. The electrode gap was kept at 180 mm. Hot water set at 60 °C was pumped. Then, the RF heating system was turned on. Time was recorded when the temperature of the test point changed from 30 to 60 °C (come up time-1, CUT-1) to find the cold spot.

To evaluate the effect of SW temperature and electrode gaps on the heating rate of the samples, different SW temperatures (116 °C, 121 °C, and 124 °C) and electrode gaps (160, 170, 180, 190, and 200 mm) were chosen. The fiber optic sensor was inserted into the cold spot of the sample in the above-mentioned experiment. The time needed for the cold spot temperature of the sample to increase from 30 °C to SW temperature (come up time-2, CUT-2) was measured. Three electrode gaps in each temperature representing the slowest, the fastest, and the moderate heating rate of the cold spot were selected to obtain the temperature-time profile by measuring every minute or less in comparison with the control (SW sterilization).

### 2.5. Inactivation of G. stearothermophilus Spores

#### 2.5.1. Spore Inoculation into PSPS

Before inoculation, the PSPS was maintained at 25 °C for 2–4 h. A spore suspension of 100 μL was injected into the PSPS sample using a 1 mL sterile injector (Shengguang medical instrument Co., Ltd., Henan, China) in a biosafety clean bench (YT-CJ-1N; Beijing Yataikelong Instrument Technology Co., Ltd., Beijing, China), and then, it was shaken for 30 min. The lid of the container was covered tightly anywhere outside the clean bench to prevent contamination. The initial concentration of *G. stearothermophilus* spores in the PSPS was approximately 10^6^ CFU/g.

#### 2.5.2. The Sterilization of RFSW

Three electrode gaps (chosen in Section 2.4) and three SW temperatures (116 °C, 121 °C, and 124 °C) were used to find appropriate parameters in sterilization that could reduce *G. stearothermophilus* spores by at least 5 log and maximize the protection of PSPS quality. The PSPS samples sterilized by SW alone served as the control (CON-SW).

#### 2.5.3. Enumeration of Surviving Cells

PSPS samples were cooled after sterilization and were placed into a sterile homogeneous bag. They were serially (1:10) diluted with 0.8% saline after blending for 2 min in a laboratory mixer (LC-08, Ningbo Licheng Instrument Co., Ltd., Ningbo, China). Then, 0.1 mL of each dilution (1 ml of blended PSPS sample) was poured into NA, spread evenly, and incubated reversely at 55 °C for 24–48 h. The minimum detection limit of this method was 1 CFU g^−1^.

### 2.6. Quality Analysis

Quality analysis was performed immediately after the sterilized sample was cooled to room temperature.

#### 2.6.1. Water Loss

Before sterilization, the boiled pork slices were weighted (*W*_0_). After sterilization, the PSPS sample was cooled to room temperature. Then, the pork slices were removed from the PSPS mixture. The surface moisture was wiped off slightly with a tissue. The sterilized pork slices (*W*_1_) were weighed. The water loss was calculated using the following Formula (1):Weight loss (%) = (*W*_0_ − *W*_1_)/*W_0_* × 100%(1)


Each treatment was replicated three times.

#### 2.6.2. Texture Measurement

The textural characteristics of pork slices sterilized by RFSW and CON-SW were measured by a texture analyzer (TA.XT Plus, Stable Micro system, Ltd., Godalming, UK). A 5 mm-diameter probe was inserted twice into the pork slices at 2 mm depth to obtain the hardness, chewiness, and resilience data. The apparatus parameters of pretest, test, and post-test speeds were set at 1 mm/s [21]. Each treatment was replicated three times.

#### 2.6.3. Measurement of Thiobarbituric Acid Reactants (TBARS)

The TBARS values of PSPS sample unsterilized and sterilized by RFSW/CON-SW, including pork slices and soup, were determined by using a method described by Liu, et al. [22] with slight modification. Specifically, 10 g sample (pork slices or soup) was mixed with 50 mL 7.5% aqueous solution of trichloroacetic acid (TCA with 0.1% of ethylene diamine tetraacetic acid) for malondialdehyde extraction. Then, the mixture was fully homogenized, shaken for 5 min, and centrifuged at 6000× *g* for 5 min. Next, 2 mL supernatant was mixed with 2 mL 0.02 moL/L thiobarbituric acid (TBA) and reacted at 100 °C in a water bath for 30 min. It was cooled quickly and centrifuged at 1600× *g* for 5 min. Finally, the supernatant was mixed with 2 mL trichloromethane (analytical reagent), shaken for 10 s, and left to stand for delamination. The absorbance of the supernatant was measured at 532 and 600 nm by a spectrophotometer (P7, Shanghai MAPADA instruments Co., Ltd., Shanghai, China). The blank solution of mixed 2 mL TBA and 2 mL TCA was used as the control. The TBARS value was expressed as mg/kg of malondialdehyde and calculated by using the following Formula (2) [22]:TBARS (mg/kg) = (*A_532_* − *A_600_*)/150 × (1/10) × 72.6 × 100(2)

Each treatment was replicated three times.

#### 2.6.4. Sensory Evaluation

Sensory evaluation of PSPS samples was carried out using quantitative descriptive analysis (QDA). Before the analysis, 10 panelists met weekly for a twelve-week period to establish descriptor attributes and terminology around attributes of PSPS. The analytical content was developed and improved through consensus and voting methods, and the evaluation process and descriptive terms with specific meanings were established through group discussions [23]. A descriptor list was developed to evaluate the attributes: appearance, odor, tenderness, juiciness, soup taste, and overall acceptability. Panelists may have given a low evaluation to the odor and taste of the sample because of its unpleasant overcooked and rancid flavor. The score of the items ranged from 1 to 5 with 1 being the lowest (inferior) and 5 being the highest (superior). The samples used for testing were reheated by a microwave before evaluation, and the microwave oven was turned off when the temperature at the cold spot reached 60 °C. Each sample was made in duplicate, and the order of the samples’ placement was randomized. Moreover, PSPS samples (50 mL soup with three pieces of meat slices) were put in special edible plastic cups. Panelists assessed the random coded samples individually in a quiet, odorless, and enclosed environment. Each sample was evaluated three times by a total of 10 panelists of sensory evaluators.

### 2.7. Microstructure Analysis

A scanning electron microscope (SEM) (Nano SEM-450, USA) was used to intuitively reflect the effects on the fiber structure of meat slices in the processing of RFSW or SW alone and observe the changes in the cross-sections in cell morphology before and after sterilization.

The samples were prepared by soaking sections (5 × 5 × 3 mm^3^) in 1.5 mL of 2% (*v*/*v*) glutaraldehyde solution at 4 °C for more than 10 h fixation. Then, the samples were rinsed with phosphate buffer (pH 7.2, 100 mmol/L) three times for 10–15 min each, dehydrated in different gradient concentration ethanol (30%, 50%, 70%, 80%, and 90% (*v*/*v*)), in turn, three times each for 30 min. After dehydration, the sample was taken out and dried in supercritical CO_2_ fluid overnight [20,21]. Finally, samples were coated in a sputter coater and photographed in SEM.

### 2.8. Statistical Analysis

All experiments were performed in triplicate and the results were expressed as mean ± standard deviation. Significant differences (*p* < 0.05) were analyzed by one-way analysis of variance (ANOVA) using SPSS software (Version 20.0, IBM Corp., Armonk, NY, USA).

## 3. Results and Discussion

### 3.1. Determination of Cold Spot

It is important to reveal how the time–temperature relationship impacts the inactivation of microorganism and the quality of a meat product [8]. Eight points in the PSPS sample were measured to discuss RFSW heating uniformity. The cold spot representing the heat limitation is meaningful for the food sterilization process.

The CUT-1 in eight points of the PSPS sample is shown in Figure 4. The central point in the middle layer (point C) was the cold spot, but it showed no significant difference (*p* > 0.05) from the two other points (points D and E) in the middle layer. The overall heating rate in the bottom layer was slower than that in the top layer. The nonuniform heating of PSPS samples resulted from the nonuniform distribution of RF energy and samples. The top, middle, and bottom layers of the samples consisted of oil, soup (salt solution), and pork slices, respectively. The RFSW heating of PSPS manifested as a margin heating pattern in this study, which was contrary to the core heating pattern identified by Jiang et al. [10]. Possibly, the PSPS sample was surrounded by SW, which transfers heat from the surface to the interior. The fastest heating rate was observed in the top layer. That was because the contact surface between edible oil and soup had a better heating efficacy base due to the fact that the electric intensification happens where two contact dielectric materials have a notable difference in dielectric constant [24]. The dielectric constant of edible oil was the smallest among all food items, whose result was similar to the research by Valantina et al. [25]. In the top layer, point B was heated faster than point A (*p* < 0.05), which possibly led to the outcome that the electric field was deflected by the edges in a sample with a cylindrical shape. Thus, point C was selected as the cold spot for the subsequent experiments.

### 3.2. The Heating Rate and Curve

The inactivation efficiency of microorganisms of a meat product depends on the temperature achieved during heat treatment, whereas the quality change depends on the time [26]. Therefore, the sterilization process needs to be built in a mild but time-saving way.

The effect of SW and electrode gap on heating rate is shown in Table 2. The average heating rate of 116 °C increased continuously with the increasing electrode gap from 160 mm to 190 mm and decreased sharply from 190 to 200 mm; the same results were found for 121 °C and 124 °C. Many studies indicated that a narrower electrode gap strengthens an electric field [11,27]. Also, a narrower electrode gap leads to worse heating uniformity when food materials were relatively smaller-sized and placed on the center of the bottom electrode of the RF cavity [28]. Consequently, the heating rate of the cold spot in the PSPS sample increased, possibly because it absorbed more RF energy as the electrode gap increased (from 160 to 190 mm). It is hard to conclude about the effect of SW temperature on heating rate. To find the appropriate condition, the electrode gaps of 190, 200, and 170 mm (representing the fastest, slowest, and moderate heating rates, respectively) in each SW temperature were selected to complete the inactivation test of *G. stearothermophilus* spores.

Temperature–time curve of PSPS at different electrode gaps at 116 °C, 121 °C, and 124 °C is shown in Figure 5.

RFSW sterilization reduced at least half of the sterilization time, thereby reaching the target temperature compared with the CON-SW treatment. A greater slope difference between two curves of RFSW and CON-SW in a temperature was observed in the back part of the curve than in the front part, thereby showing that the electrode gap significantly increased the heating rate of the latter part of the heating process. As the temperature difference between PSPS and SW decreased, the mainstay of heat transfer impetus changed from temperature difference to RF energy. This conclusion can be supported by another phenomenon that the heating rate difference between RFSW-190 and RFSW-170 or RFSW-200 was more obvious at 116 °C and 121 °C than at 124 °C in the front part of the heating process. At 124 °C, a greater temperature difference was found between SW and PSPS samples. We interestingly found that 121 °C-30 min SW sterilization was not enough for the PSPS sample to reach an internal temperature of 121 °C, even though this is widely used in the food industry.

### 3.3. Inactivation of G. stearothermophilus spores

*G. stearothermophilus* spores, which have extremely strong heat resistance, are known as an indicator species in thermal food processing and used as a substitute for *Clostridium botulinum* to address biosafety concerns [18]. The inactivation of *G. stearothermophilus* spores in PSPS by RFSW sterilization at different temperatures and electrode gaps is shown in Figure 6. The SW temperature and electrode gap significantly affected the inactivation of *G. stearothermophilus* spores. The shortest time of 5 log reduction in *G. stearothermophilus* spores was observed in RFSW-190 /124 °C sterilization for 12 min, whereas the CON-SW/116 sterilization had the longest time of 36 min. Under each SW temperature, RFSW-190 sterilization showed the fastest spore reduction rate, whereas RFSW-200 had the slowest. In accordance with the heating rate results in Section 3.2, it could be concluded that the efficiency of *G. stearothermophilus* spores inactivation was positively correlated with heating rate. When the SW temperature increased from 116 to 121 °C and 124 °C, less time was needed to reach the 5 log reduction in *G. stearothermophilus* spores under a certain electrode gap, indicating that temperature was an important factor in RFSW sterilization. Whether higher temperature of thermal treatment and shorter time were more desirable for food sterilization has not been determined. To protect the edible quality and nutrients, conducting a physicochemical property test was necessary. Therefore, a sample at the end of each sterilization process was selected for subsequent quality measurement.

From the comparison and detailed analysis of the time–temperature profile and *G. stearothermophilus* spores inactivation curve, more valuable information could be obtained. Under 121 °C, the number of surviving spores sterilized by RFSW increased at 6 min compared with 3 min, indicating that *G. stearothermophilus* spores may germinate in soup and pork slices from 3 min to 6 min. At 6 min, according to Figure 5, the temperature of the cold spot reached around 105 °C, which seems insufficient for *G. stearothermophilus* spore activation, according to the research of Finley et al. [29]. Some nutrients with small molecular weight in the PSPS (e.g., amino acids and sugars) help accelerate spore germination [30]. Moreover, a higher temperature was reached in some points except for the cold spot. The time–temperature profile and the inactivation curve showed great correspondence.

The holding times (time above the target temperature) at 124 °C (0.9 min at 190 mm, 0 min at 170 mm, and −2.25 min at 200 mm) and 121 °C (2.81 min at 190 mm, 2.46 min at 170 mm, and 1.39 min at 200 mm) showed a slower heating rate corresponding to shorter holding time, but at 116 °C (8.44 min at 190 mm, 9.65 min at 170 mm, and 11.3 min at 200 mm), the opposite result was observed. SW temperature influenced the sterilization efficiency in some way, perhaps in relation to the heating cumulative effect [29,30]. Moreover, a larger slope can be observed in the tail of most curves, indicating that a critical temperature varied the sterilization efficiency.

### 3.4. Quality Evaluation

#### 3.4.1. Water Loss

The mixture of potato starch and egg white gelatinized and formed a gel covering the surface of the pork slices at high temperature, thereby somehow preventing water loss. Nonetheless, the electrode gap in each SW temperature still affected the water loss markedly according to Table 3. Pork slices sterilized by RFSW-200 and RFSW-170 had more significant (*p* < 0.05) smaller water loss than those sterilized by RFSW-190 and CON-SW at 116 °C. At 121 °C, a significant difference was observed between CON-SW and RFSW sterilization (*p* > 0.05), whereas no significant difference was found at 124 °C. Prolonged heating time and elevated temperature are the two main factors affecting water loss [26], in which a too-fast heating rate as well as longer heating time both could lead to severe moisture drop, because water expulsion happens when muscle fiber shrinks and sarcomere length decreases [26,31]. This finding possibly accounted for the fact that CON-SW and RFSW-190 sterilization resulted in higher water loss. CON-SW sterilization more rapidly reached 5 log *G. stearothermophilus* spores reduction at 124 °C than at 121 °C and 116 °C (18, 24, and 36 min, respectively), so that less time was allowed to accumulate the water loss difference between CON-SW and RFSW treatment, thereby causing a smaller structural change. RFSW-170 in each temperature was shown to have a better balance between heating time and temperature and consequently led to the best condition for RFSW to reduce water loss.

#### 3.4.2. Texture Measurement

The texture of samples sterilized by RFSW and CON-SW under each temperature is shown in Table 3. The hardness and chewiness of samples sterilized by RFSW-170 and RFSW-200 were significantly smaller (*p* < 0.05) than those sterilized by RFSW-190 and CON-SW, possibly because RFSW-190 reached a higher end temperature for its fastest heating rate, and CON-SW needed a long duration of time to achieve a 5 log reduction of in *G. stearothermophilus* spores. Three components primarily determined the hardness of the meat, namely, myofibrillar proteins, connective tissues, and collagen. The secondary, tertiary, and quaternary structures of these protein-based substances can be affected differently by different heat intensities. A too-fast heating rate of the sample induced denaturing, the connective tissue and collagen dissolving, and myofiber shrinkage, and longer heating time of sample led to severe protein denaturation as well, with both increasing the hardness of the meat [31]. Moreover, according to the results of water loss, RFSW-190 caused greater water loss than RFSW-170 and RFSW-200 and gave results closer to CON-SW, which also led to severe protein denaturation. During the heating, structure-related protein starts to denature, the connective tissue and collagen dissolve, and myofibers shrink, thereby increasing the toughness of the meat [7]. Palka and Daun [32] indicated that compared with lower temperature, a closer myofiber fiber gap was observed at 121 °C, thereby showing that a higher temperature leads to more severe myofiber shrinkage. In terms of resilience, sample resilience under sterilization by RFSW-170 was significantly higher (*p* < 0.05) than that under sterilization by RFSW-190, RFSW-200, and CON-SW at 121 °C and 124 °C, whereas no significant difference (*p* > 0.05) was found at 116 °C. The resilience of meat might be related to the swelling of fiber, as reflected by the diameter of the fiber. RFSW-170 reached 5 log *G. stearothermophilus* spores reduction under moderate heating, thereby preventing the excessive increase in myofiber diameter.

#### 3.4.3. Oxidation Analysis

Under the required cooking or sterilization intensity, a lower temperature and a shorter time are desired to reduce oxidation of meat products in industry. Strong heat would induce the oxidization of polyunsaturated fatty acids, thereby yielding aldehydes, free fatty acids, and other substances. This would cause quality failure, i.e., off flavor, which may lead to safety problems and reduce consumer acceptability [33,34].

The influence of SW temperature and electrode gap on the TBARS value of pork slices and soup is shown in Figure 7. TBARS value reflects the secondary oxidation degree of food. Apparently, oxidation in soup sterilized by RFSW or CON-SW was greater than in pork slices. Soup in the top layer of the PSPS sample reached a higher temperature according to the result in Figure 4. Autoxidation became more intense at higher temperature [35]. Soup contains more fatty acids and transfers heat faster compared with pork slices. The PSPS sample sterilized by RFSW-170 produced significantly (*p* < 0.05) less malonaldehyde than that sterilized by RFSW-190 at 116 °C and 121 °C. At 124 °C, no significant difference was observed among samples under all sterilization conditions. These phenomena indicated that electrode gap influenced the extent of oxidation less under high temperature or over a long time.

To summarize, the 170 mm electrode gap was the best condition to reduce heating time and protect PSPS quality. Thus, RFSW-170 was selected to complete the sensory evaluation.

#### 3.4.4. Sensory Evaluation

The sensory-evaluation results of samples sterilized by RFSW-170 in each temperature and CON-SW are presented in Figure 8. Significant differences (*p* < 0.05) were observed in appearance, odor, soup taste, and overall acceptability between RFSW and CON-SW, whereas no significant differences (*p* > 0.05) were found in tenderness and juiciness. The scores of each item under RFSW at three temperatures showed no significant differences (*p* > 0.05), indicating that the panelists could hardly tell the distinction among PSPSs.

Continuous heating promotes oxidation in pork slices and lipid oxidation and hydrolysis in the edible oil of the soup, leading to off-flavor generation and the loss of sensory quality [36,37]. The panelists preferred the odor and soup taste of RFSW-sterilized samples compared with the CON-SW group, which might be due to the lower level of TBARS (Section 3.4.3) in meat sterilized by RFSW-170.

#### 3.4.5. Microstructure Analysis on SEM Imaging

Changes in the morphological myofibrillar structure of sliced pork in SEM images subjected to RFSW and conventional SW sterilization are shown in Figure 9. Figure 9a demonstrates the internal microstructure of raw pork slices, which clearly showed the sparse arrangement of myofibrils with clearly visible gaps. When samples were heated to 116 °C by SW (Figure 9b) alone, the myofibrils of the pork slices were arranged more compactly, contrasting with control. Similar phenomena can be found in Figure 9c–e when samples were treated by RFSW (116 °C) at the fixed electrodes of 170 mm, 190 mm, and 200 mm, respectively. This phenomenon may be attributed to denaturation of the proteins that make up the myofibrillar and drainage of interfiber water [38].

When the temperature rose to 121 °C, the peptide bonds were hydrolyzed and denatured, and collagen cross-links were broken; meanwhile, the hardness of the pork slices decreased (Figure 9f–i). Similarly, more significant results can be found in Figure 9j–m when samples were treated at the SW temperature of 124 °C. Li et al. [39] revealed that the structure of chicken muscle fibers was packed more tightly after the chicken was heated from 45 to 95 °C at the center temperature. As a consequence, the myofibrillar structure of sliced pork heated by RFSW heating at a fixed electrode of 170 mm showed less variation compared to the control because the samples were heated at a relatively suitable heating rate under this condition, which made its heating rate faster than that with SW heating without reaching excessive temperature in a short period of time. Moreover, a faster heating rate resulted in less structural change in myofibrils, reducing the water loss of the pork slices and decreasing the hardness of the pork slices, which corresponded to the results above in Section 3.4.1 and Section 3.4.2. Thus, revealing the microstructural changes in meat during the process of sterilization is necessary.

## 4. Conclusions

A new device, which can apply RF energy and SW simultaneously, was employed in this study to eliminate microorganisms inoculated in RTE PSPS. The availability of the test results proved that the application was stable and serviceable. The cold spot lay in the geometric center of the Teflon container, but we were surprised by the difference in heating rate between the upper and bottom layers of the PSPS sample. Such a difference shows that the shape and properties of the material had a great impact on the electric field distribution and heating rate. A deeper study of RF heating complex food materials will pose a greater challenge. Less time was needed for the cold spot in PSPS to reach the SW temperature when RFSW sterilization was applied compared with when SW sterilization was applied. Also, less time was required for the PSPS sample to reach a 5 log reduction in *G. stearothermophilus* spores under RFSW sterilization. It was found that RFSW-190, despite its fastest heating rate, resulted in even worse quality compared to the conventional method. On the contrary, RFSW-170 sterilization overall reduced the water loss and thermal damage to texture, oxidation, tissues, and cells of the PSPS sample and kept better sensory properties. Thus, RFSW-170 was more suitable for sterilization in this study. It is hard to reach a conclusion about the effect of SW temperature on the quality of PSPS samples, but RF energy affected it less at 124 °C.

Although we achieved some desirable results in terms of heating rate and quality protection, it is still unclear whether this new RTE food sterilization technology saves more energy, which is an important issue to be explored in subsequent research. This study suggested considerable promise for the application of RFSW to RTE food sterilization.

## Figures and Tables

**Figure 1 foods-11-02841-f001:**
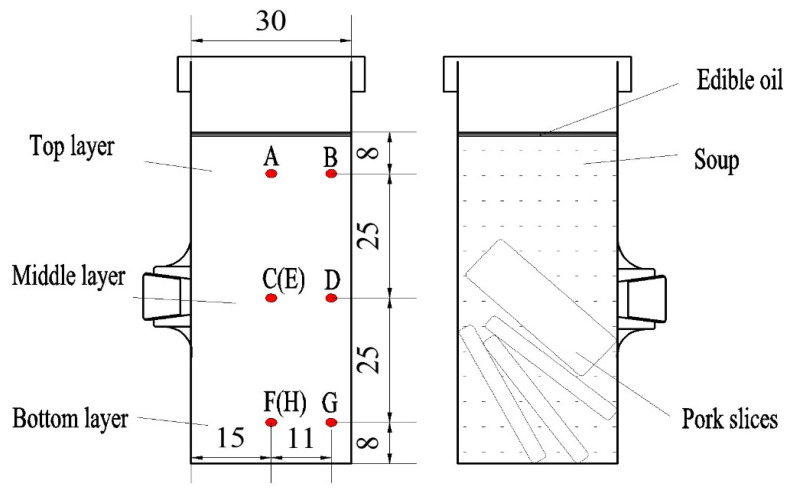
Position of fiber optic sensor in the Teflon container and distribution of the poached spicy pork slice sample in the Teflon container (all dimensions are in mm). Points A, B, C, D, F, and G were used to measure the soup temperature, whereas E and H were used for the pork slices’ internal region.

**Figure 2 foods-11-02841-f002:**
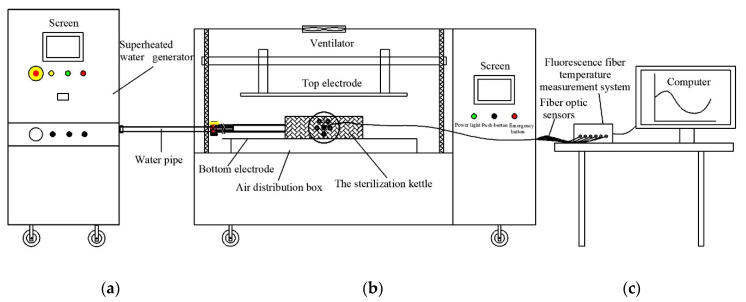
Simplified schematic diagram of RFSW heating system including superheated water generator (**a**), radio frequency heating system (**b**), and fiber optic temperature measurement system (**c**).

**Figure 3 foods-11-02841-f003:**
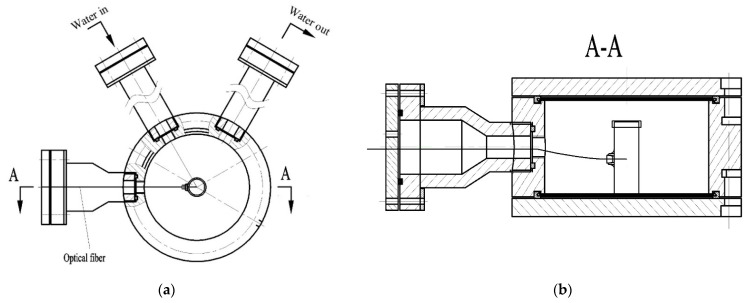
Schematic of the sterilization kettle (**a**) and partial section view of the sterilization kettle (**b**).

**Figure 4 foods-11-02841-f004:**
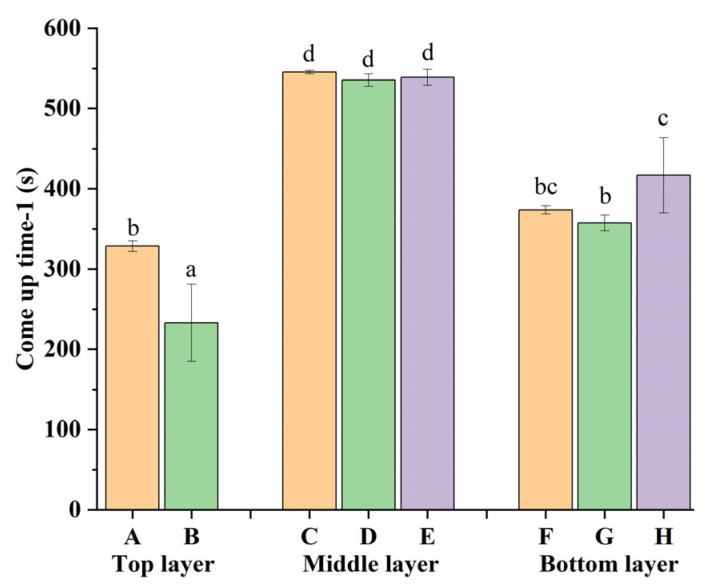
Come-up time-1 from 30 °C to 60 °C under 180 mm electrode gap at different points of the PSPS sample. Six points (A, B, C, D, F, and G) were in soup and two points (E and H) were in pork slices. (Different lowercase letters (a–d) indicate a significant difference (*p* < 0.05)).

**Figure 5 foods-11-02841-f005:**
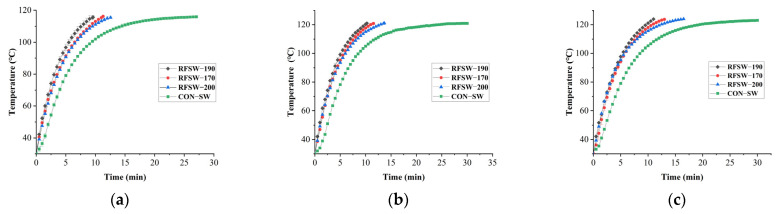
Time–temperature profiles of point C in poached spicy pork slices in different electrode gaps at 116 °C (**a**), 121 °C (**b**), and 124 °C (**c**). RFSW means radio frequency combined superheated water at an electrode gap; CON-SW means superheated-water sterilization without radio frequency energy.

**Figure 6 foods-11-02841-f006:**
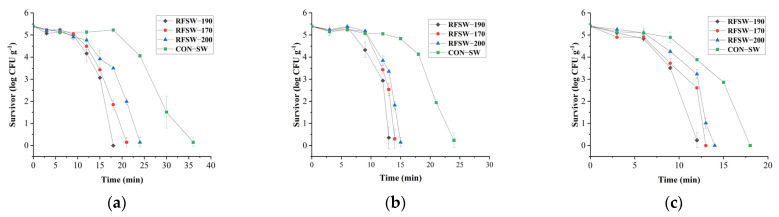
Log survivor of *G. stearothermophilus* spores in PSPS under the different electrode gaps and SW temperatures of 116 °C (**a**), 121 °C (**b**), and 124 °C (**c**).

**Figure 7 foods-11-02841-f007:**
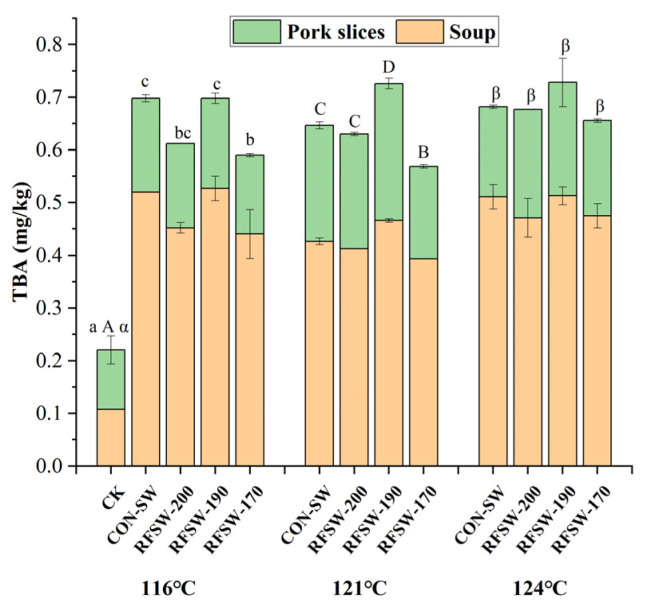
Thiobarbituric reactive substances of poached spicy pork slices sample sterilized by RFSW under different electrode gaps and superheated water temperatures. Different lowercase letters (a–f, A–D, and α,β) indicate significant differences (*p* < 0.05) within the different temperatures of 116 °C, 121 °C, and 124 °C. CK means unsterilized poached spicy pork slices; RFSW means radio frequency combined superheated water at a certain electrode gap; CON-SW means superheated water sterilization alone.

**Figure 8 foods-11-02841-f008:**
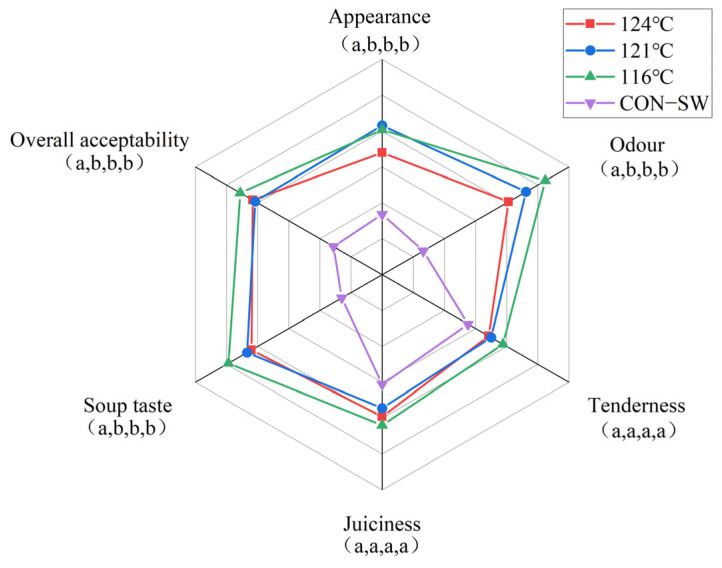
Sensory evaluation of poached spicy pork slice samples sterilized by radio frequency combined superheated water in three temperatures and superheated water alone in 116 °C. The four letters in brackets belong to the four conditions from the center to the outside. Different lowercase letters (a–b) indicate a significant difference (*p* < 0.05) within the same object.

**Figure 9 foods-11-02841-f009:**
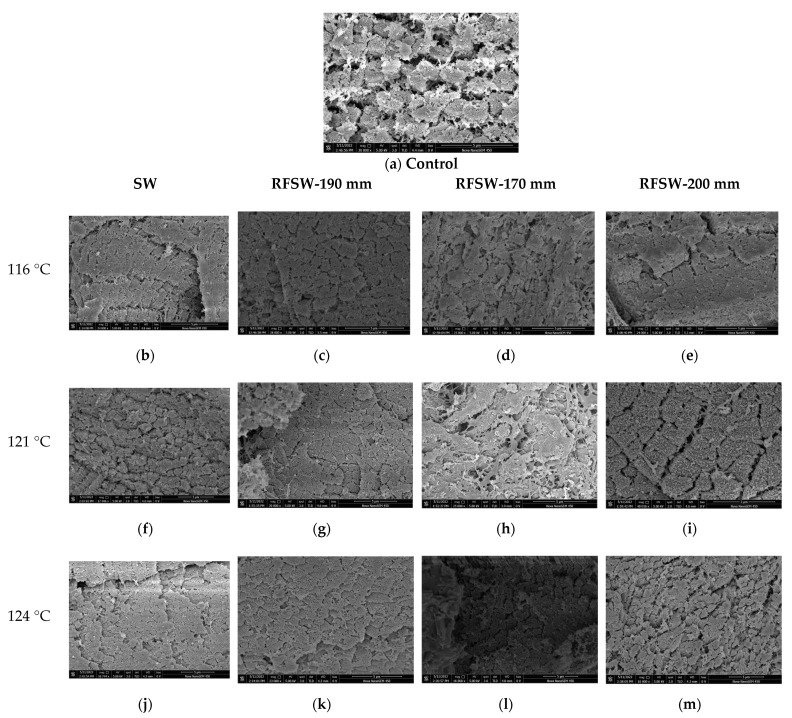
SEM micrographs of myofibrils in pork slices after sterilization with different electrode gaps and superheated-water temperature (control group means starching pork slice without sterilization).

**Table 1 foods-11-02841-t001:** Materials and dosage involved in PSPS cooking (listed in the order of use).

Soup			Pork Slices		
	Dosage (g)	Information		Dosage (g)	Information
Spring onion	15.0 ± 0.1	Hao&Duo supermarket (Used stem only)	Salt	1.50 ± 0.05	Hao&Duo supermarket
Ginger	5.0 ± 0.1	Hao&Duo supermarket (Removed the peel)	Chinese rice wine	2.0 ± 0.1	Jingzhiliaojiu, Beijing Ershang Wangzhihe Food Co., Ltd., Beijing, China
Garlic	7.0 ± 0.2	Hao&Duo supermarket (Removed the peel)	Egg white	9.0 ± 0.5	Hao&Duo supermarket
Bean paste	15.0 ± 0.1	Pixian Bean paste, Pixian Star Seasoning Co., Ltd., Pixian, China	Potato starch	9.0 ± 0.5	Xi’an Panfeng Shunda Trading Co., Ltd., Xi’an, China
Salt	4.0 ± 0.1	Hao&Duo supermarket			
Chicken extract	0.40 ± 0.05	Hao&Duo supermarket			
Sugar	4.0 ± 0.1	Hao&Duo supermarket			
Light soy sauce	2.0 ± 0.1	Haday soy sauce, Foshan Haitian Condiment Food Co., Ltd., Foshan, China			
Chinese rice wine	2.0 ± 0.1	Jingzhiliaojiu, Beijing Ershang Wangzhihe Food Co., Ltd., Beijing, China			
Water	400 ± 1	Laboratory drinking water (Food grade)			

**Table 2 foods-11-02841-t002:** Come-up time and heating rate from 30 °C to superheated water set temperature at different electrode gaps.

Superheated Water Temperature (°C)	Electrode Gaps (mm)	Come-Up Time-2 (min)	Heating Rate (°C/min)
116	160	12.09 ± 0.52	7.12
	170	11.35 ± 0.03	7.58
	180	10.26 ± 0.05	8.38
	190	9.57 ± 0.87	8.99
	200	12.70 ± 0.81	6.77
121	160	13.25 ± 0.07	6.87
	170	11.54 ± 0.48	7.89
	180	10.63 ± 0.88	8.56
	190	10.19 ± 0.51	8.93
	200	13.61 ± 0.56	6.69
124	160	13.54 ± 0.13	6.94
	170	13.05 ± 0.41	7.23
	180	11.62 ± 0.14	8.10
	190	11.12 ± 0.14	8.47
	200	16.43 ± 0.11	5.72

**Table 3 foods-11-02841-t003:** Water loss and texture of PSPS sample sterilized by radio frequency combined superheated water in different electrode gaps and at different superheated water temperatures. RFSW means radio frequency combined superheated water at an electrode gap; CON-SW means superheated water sterilization alone.

Sterilization	Water Loss (%)		Texture		
			Hardness (g)	Chewiness	Resilience
116 °C	RFSW-200	8.47 ± 0.88 ab	157.63 ± 15.22 ab	41.29 ± 2.82 a	0.181 ± 0.003 ab
	RFSW-190	18.15 ± 2.14 f	324.97 ± 3.68 h	158.85 ± 2.12 h	0.162 ± 0.003 a
	RFSW-170	7.91 ± 0.37 a	161.97 ± 9.23 ab	43.18 ± 6.13 ab	0.181 ± 0.001 ab
	CON-SW	16.51 ± 0.62 f	264.51 ± 5.11 f	70.94 ± 3.21 c	0.161 ± 0.002 a
121 °C	RFSW-200	11.80 ± 0.97 cde	213.06 ± 11.26 e	50.32 ± 1.83 ab	0.174 ± 0.001 a
	RFSW-190	13.87 ± 0.67 e	249.26 ± 0.54 f	99.85 ± 5.59 e	0.162 ± 0.001 a
	RFSW-170	10.45 ± 1.12 bc	181.13 ± 5.79 cd	53.86 ± 2.47 b	0.274 ± 0.004 d
	CON-SW	17.71 ± 0.84 f	307.32 ± 8.84 g	102.05 ± 5.93 e	0.163 ± 0.003 a
124 °C	RFSW-200	11.74 ± 0.34 cde	168.79 ± 1.01 bc	107.99 ± 12.39 e	0.232 ± 0.033 c
	RFSW-190	12.85 ± 0.84 de	190.41 ± 10.1 d	133.25 ± 8.92 f	0.204 ± 0.001 b
	RFSW-170	9.56 ± 0.82 ab	145.61 ± 7.21 a	83.89 ± 4.19 d	0.307 ± 0.032 e
	CON-SW	10.51 ± 2.62 bcd	228.37 ± 5.28 e	144.24 ± 4.35 g	0.242 ± 0.006 c

Different lowercase letters (a–f) indicate a significant difference (*p* < 0.05) within the same column. RFSW-radio frequency combined superheated water in an electrode gap; CON-SW-superheated water treatment without radio frequency energy.

## Data Availability

Data on this study are available in the article.

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
