# Peer review of "Sterilizing Ready-to-Eat Poached Spicy Pork Slices Using a New Device: Combined Radio Frequency Energy and Superheated Water"

_foods, 2022, doi:10.3390/foods11182841_

Round 1

Reviewer 1 Report

The submitted paper's content is compatible with the relevant special issue. The aim of the study is presented clearly. The experimental setup, procedure, and design are appropriate to achieve the goal. The study's experimental results can potentially contribute to the literature. There are some queries and problems that should be explained, discussed, and/or corrected in the manuscript to enhance the quality of the study:

1.     All the nominal values and their standard deviations should be presented with the same number of digits in Tables 2 and 3.

2.     It was mentioned that RFSW-190 and CON-SW are not similar in terms of heating rate or processing times. However, RFSW-190 caused greater water loss than RFSW-170 and RFSW-200 and gave results closer to CON-SW. What could be the reason for this situation?

3.     Similarly, the textural properties of RFSW-190 were similar to CON-SW. Please, discuss the reason for this situation with the literature studies.

4.     Lines 421-423: It is not clear how the authors made this selection. For instance, RFSW-170 did not provide the fastest temperature rise. Which criteria were used and which statistical method was applied for this decision-making process?

Author Response

Response to Reviewer 1 Comments

Dear,

Thank you for your careful reconsideration of our manuscript. I really appreciate all the comments and suggestions! We have studied the comments carefully and attempted to address all of the raised issues. These comments are helpful for improving our paper. The response to the comments are as following.

Point 1: All the nominal values and their standard deviations should be presented with the same number of digits in Tables 2 and 3.

Response 1: Thank you for pointing this out. We have modified the values in Tables 2 and 3 ( line 313-314 and 397-398)

Point 2: It was mentioned that RFSW-190 and CON-SW are not similar in terms of heating rate or processing times. However, RFSW-190 caused greater water loss than RFSW-170 and RFSW-200 and gave results closer to CON-SW. What could be the reason for this situation?

Response 2: Prolonged heating time and elevated temperature are the two main factors affecting it water loss, in which too fast heating rate as well as longer heating time both can lead to severe moisture drop for the reason of muscle fiber shrinks and sarcomere length decreases when water expulsion happens as described at lines 380 to 383 in 3.4.1. According to the results in Table 3, RFSW-190 caused greater water loss than RFSW-170 and RFSW-200 and gave results closer to CON-SW, which means only by choosing a milder heating method could achieved the ideal result of water loss. Overall, according to the results of inactivation of G. stearothermophilus in 3.3, the better process condition was provided by RFSW-170.

Point 3: Similarly, the textural properties of RFSW-190 were similar to CON-SW. Please, discuss the reason for this situation with the literature studies.

Response 3:  Texture measurement is similar to the above results in Point 2. The hardness of meat is mainly determined by three components, namely myofibrillar proteins, connective tissues, and collagen,whose secondary, tertiary, and quaternary structures can be affected differently by different heat intensities as discussed in 3.4.2 (lines 410-415). Too fast heating rate of sample induced the denature, the connective tissue and collagen dissolve, and myofiber shrink, and longer heating time of sample can lead to severe protein denaturation as well, which bothe increased the hardness of the meat. Besides, according to the results in water loss, RFSW-190 caused greater water loss than RFSW-170 and RFSW-200 and gave results closer to CON-SW, which also lead to severe protein denaturation. Thus, CON-SW has similar texture characteristics with RFSW-190 treatment group in terms of hardness and chewiness. Considering the taste, water loss and sterilization of PSPS, the better process condition was provided by RFSW-170.

Point 4: Lines 421-423: It is not clear how the authors made this selection. For instance, RFSW-170 did not provide the fastest temperature rise. Which criteria were used and which statistical method was applied for this decision-making process?

Response 4: According to the results of this study, it was found that RFSW-190, despite its fastest heating rate, result in even worse quality compared to conventional method. On the contrary, RFSW-170 sterilization overally reduced the water loss and thermal damage on texture ( including better performance in hardness, chewiness and resilience ), oxidation, tissues and cells of PSPS sample and kept better sensory properties. A milder heating method of RFSW-170 could achieved the ideal goals of this research

Reviewer 2 Report

ID: foods-1896838

TITLE: Sterilizing ready-to-eat Poached Spicy Pork Slices using a new device: combined radio frequency energy and superheated water 

A new device was used to inactivate G. stearothermophilus spore in Poached Spicy Pork Slices. Radiofrequency energy and superheated water were applied simultaneously. The effects of electrode gap and Superheated water temperature on heating rate, spore inactivation, physiochemical properties (water loss, texture, and oxidation), sensory properties, and SEM of samples were investigated. The investigation is new and could significantly impact the area of research. It presented results that are fine discussed and well interconnected among different analyses. The minor comments I have about it is the following.

In Figure 7 is confusing the simbols used to show significant statistical differences.

In Figures 1 and 7, de error bars are not shown correctly.

Author Response

Response to Reviewer 2 Comments

Dear,

Thank you for your careful reconsideration of our manuscript. I really appreciate all the comments and suggestions! We have studied the comments carefully and attempted to address all of the raised issues. These comments are helpful for improving our paper. The response to the comments are as following.

Point 1: In Figure 7 is confusing the simbols used to show significant statistical differences.

Response 1: Figure 7 showed the thiobarbituric reactive substances of poached spicy pork slices sample sterilized by RFSW under different electrode gaps. In order to make the results more comparable among three different superheated water temperatures, we put all the data in a figure. In this figure, CK means unsterilized poached spicy pork slices; RFSW- means radio frequency combined super-heated water at a certain electrode gap and CON-SW means superheated water sterilization alone. Different lowercase letters (a-f) indicate a significant difference (p < 0.05) within the temperature of 116 °C, different lowercase letters (A-D) indicate a significant difference (p < 0.05) within the temperature of 121 °C, and different lowercase letters (α-β) indicate a significant difference (p < 0.05) within the temperature of 124 °C. CK as a common control group in this part.

Point 2: In Figures 1 and 7, de error bars are not shown correctly.

Response 2: Thank you for pointing this out. You may mean de error bars are not shown correctly in Figures 4 and 7. We have corrected it at lines of 278 and 426.

Reviewer 3 Report

This research presents a novel approach for sterilisation of ready-to-eat products, in particular, poached spicy pork slices, combining radio frequency energy and superheated water. The article is well-written, and the results are clearly presented, matching the objectives described at the beginning.

However, it lacks some detail regarding the methodology for the sensory analysis. Some minor comments are detailed below.

RET? Please review, suggested RTE (ready-to-eat)

The experimental design is misses how many replicates per treatment?

L58: salmonella in italics??

Table 1: please align "information" column, it is not clear to which ingredients refers.

L202: replace Britain by UK

Section 2.6.2 How many samples were analysed? Please specify the n.

Section 2.6.4 Please explain in detail the procedure for presenting the samples to the panellists, and how the temperature at the moment of intake was monitored, to ensure it did not affect sensory perception. Which samples were used during the training session?

Commercial samples? Was a QDA agreed first for the control sample CON-SW?

How many training sessions were needed until the panel was robust and consistent?

L310 how would the authors explain this phenomena, how this finding affects the results, when the standard method was considered 121ºC- 30 min?

Table 3: include n (number of replicates)

Figure 7: include n (number of replicates)

Figure 8: Caption does not seem to match the figure showing the spider charts from the sensory analysis.

L458: period after "Besides,"

Author Response

Response to Reviewer 3 Comments

Dear,

Thank you for your careful reconsideration of our manuscript. I really appreciate all the comments and suggestions! We have studied the comments carefully and attempted to address all of the raised issues. These comments are helpful for improving our paper. The response to the comments are as following.

Point 1: RET? Please review, suggested RTE (ready-to-eat)

Response 1: Thank you for pointing this out. We have corrected RET into RTE.

Point 2: The experimental design is misses how many replicates per treatment?

Response 2: We have presented that all experiments were performed in triplicate and the results were expressed as mean ± standard deviation as described in 2.8.

Point 3: L58: salmonella in italics??

Response 3: Thank you for your suggestion. We have corrected it at line 58.

Point 4: Table 1:please align "information" column, it is not clear to which ingredients refers.

Response 4: Thank you for pointing this out. We have modified it in Table 1 at line 98.

Point 5: L202:replace Britain by UK

Response 5: Thank you for pointing this out. We have replaced Britain by UK.

Point 6: Section 2.6.2 How many samples were analysed? Please specify the n.

Response 6: Thank you for your suggestion. We have added the more details at line 209.

Point 7: Section 2.6.4 Please explain in detail the procedure for presenting the samples to the panellists, and how the temperature at the moment of intake was monitored, to ensure it did not affect sensory perception. Which samples were used during the training session?

Response 7: Thank you for your suggestion. We have added the more relative details in section 2.6.4 at lines (228-244). We think it is much clear about the issue you mentioned.

Point 8: Commercial samples? Was a QDA agreed first for the control sample CON-SW?

Response 8: No, they are not commercial samples, and QDA was agreed first for the control sample CON-SW.

Point 9: How many training sessions were needed until the panel was robust and consistent?

Response 9: Thank you for your comments. We have added the more relative details in section 2.6.4 at lines 228 to 234.

Point 10: L310 how would the authors explain this phenomena, how this finding affects the results, when the standard method was considered 121ºC- 30 min?

Response 10: It is almost hardly for the internal temperature of materials to reach the target temperature (121 °C) when heating solid and semi-solid samples by the standard sterilization method of 121 °C-30min. In order to reach the commercial sterilization condition, the heating time must be extended or the target heating temperature must be increased, which will inevitably affect the relevant quality of the samples and increase energy consumption. According to the results of this study, The interior and exteriorof samples could be heated at the same time when subjected to RFSW heating, resulting that about half the processing time was saved and 5 log reduction of G. stearothermophilus spore was achieved. RFSW sterilization under 170 mm electrode gap reduced the water loss, thermal damage of texture, oxidation, tissues and cells of sample and kept a better sensory. This fully shows that RFSW sterilization owns great potential in solid or semisolid food processing engineering compare to traditional thermal sterilization.

Point 11: Table 3: include n (number of replicates)

Response 11: Thank you for your suggestion. We have added the more details at line 203, and modified the Table 3 at line 397.

Point 12: Figure 7 include n (number of replicates)

Response 12: Thank you for your suggestion. We have added the more details at line xxx, and modified the Figure 7 at line 426

Point 13: Caption does not seem to match the figure showing the spider charts from the sensory analysis.

Response 13: Thank you for your reminder. We have recorrected it at lines 428-433

Point 14: L458: period after "Besides,"

Response 14: Thank you for your reminder. We have modified it at line 492
